# Enhancing Parental Understanding of Emotions in Children with Developmental Language Disorder: An Online Parent-Led Intervention Program

**DOI:** 10.3390/healthcare12161571

**Published:** 2024-08-08

**Authors:** Fatma Canan Durgungoz, Michelle C. St Clair

**Affiliations:** 1Department of Special Education, Mersin University, 33110 Mersin, Türkiye; 2Department of Psychology, University of Bath, Bath BA2 7AY, UK; mcsc21@bath.ac.uk

**Keywords:** developmental language disorder, emotion recognition, intervention, parent-led

## Abstract

Many children with developmental language disorder (DLD) have emotion recognition and regulation difficulties, but there are currently no known interventions enhancing emotional awareness in this population. This study explores the impact of parents’ perspectives regarding children with DLD emotional understanding through a parent-led online emotion recognition (ER) intervention. Ten parents of children with DLD aged 6–11 participated in the study. A nonconcurrent multiple baseline design was employed, allowing for a rigorous analysis of changes in parental beliefs over time. Weekly data were collected through the Parents’ Beliefs About Children’s Emotions Questionnaire. Interviews were also conducted to gain deeper insights into parents’ perceptions regarding the ER skills of their children. Results indicated that parents’ beliefs about the need for guiding and supporting their child’s ER skills increased over the intervention. Interviews also supported this, and three main themes were generated. The intervention program increased parents’ awareness of (a) the importance of ER for children with DLD, (b) emotion-focused communication and engagement with their child, and (c) the integration of emotions into daily life. This study is the first known study that explores parents’ beliefs about children with DLD ER skills, highlighting the importance of supporting parents through accessible interventions.

## 1. Introduction

Recent research has highlighted that developmental language disorder (DLD) is not only characterized by language difficulties but also by difficulties in recognizing and expressing emotions effectively. Children with DLD may struggle to perceive and interpret the emotions of others accurately [1] as well as experience difficulties processing their own emotions [2], making it challenging for them to engage in meaningful social exchanges. This can lead to feelings of isolation and frustration, as they may find it challenging to connect with their peers [3] or respond appropriately to social cues [4]. Griffiths et al. [5] conducted a cohort study with children with DLD and typically developing children and suggested that children with DLD have deficits in emotion recognition (ER) skills such as recognizing facial and vocal emotion cues. Additionally, the inability to express their own emotions adequately can lead to increased frustration and emotional distress for children with DLD [6]. When they cannot effectively communicate their feelings, it may manifest as behavioral issues or emotional outbursts, which can further isolate them from their peers and potentially lead to social withdrawal [3].

While, traditionally, professionals may have prioritized the improvement of language skills as the primary focus for children with DLD, the difficulties and needs of children with DLD highlight the importance of exploring a more holistic approach. Regarding the overall development of children with DLD, professionals should also focus on interventions that support emotional understanding and expression. By addressing these emotional challenges as early as possible, professionals can potentially prevent some of the stress and emotional distress experienced by children with DLD, thereby reducing the likelihood of behavioral issues and social withdrawal and fostering more meaningful connections with peers.

### 1.1. Parents’ Thoughts about ER Abilities of Their Child

There are two parental beliefs regarding emotions: the belief that children’s emotions hold value and the belief that children’s emotions pose risks [7]. Evaluating both beliefs is crucial as they can operate in opposing ways, function independently, or align in similar directions. Parents who are aware of the significance of their children’s emotions are more likely to use the expression of both positive and negative emotions and actively engage with them during emotional moments [7]. Parents who view children’s emotions as problematic or dangerous believe that emotions are generally unhelpful and should be avoided [8]. They are more likely to discourage their children from experiencing negative emotions such as sadness, anger, and fear as they believe these emotions may cause problems. Moreover, these parents may suppress their own emotions, show less emotional expression overall, and provide limited explanations about emotions to their children [7]. There is no doubt that parental guidance and the use of emotional expressions play a pivotal role in shaping their and their child’s emotional abilities.

Parents’ perceptions of their child’s ER skills and awareness about their child’s emotional abilities can significantly impact various aspects of their parenting practices and their child’s socio-emotional development [9]. Parents who perceive their child as good at recognizing emotions may not support and guide their child to discuss their feelings openly. In contrast, parents who believe their child struggles with ER may provide additional support and guidance to help their child navigate social and emotional challenges. A parent-rated assessment indicated that there was a significant contrast between children with DLD and typically developing children in terms of the social–emotional development of their child [10]. Parent-rated assessments showed that children with DLD had lower skills in terms of social–emotional and language abilities [10]. Thus, understanding parental perceptions of their child’s emotional skills and their awareness of their child’s emotional abilities is crucial for informing parenting practices and supporting socio-emotional development.

### 1.2. The Need for Parent-Led Interventions

Studies highlighted that parents’ first and most important role is being “teachers” [11], “educators” [12] for their children, so supporting parents as active “co-participants” [13] can help them to meet their children’s unique needs. By providing parents with guidance and resources, it would be possible to enhance children’s educational experiences and support their overall development at home. For instance, Te Kaat-van den Os et al. [14] conducted a systematic review focusing on parent-led language interventions for children with developmental delay and concluded that parent-led intervention programs focused on improving communication and language skills in children with developmental disabilities seem to help children communicate better. Moreover, studies highlighted that parent-led intervention might be a cost- and time-effective approach to help families [15]. As far as we are aware, there are no studies that aimed to increase parents’ belief and understanding about supporting their DLD-affected children with perceiving and processing emotions. Thus, this study aims to investigate how a parent-led online ER intervention had an impact on parent’s beliefs about their child’s emotions.

## 2. Materials and Methods

### 2.1. Participants

To participate in this study, the inclusion criteria were as follows: (1) being a parent of a child aged 6–11 years, (2) the child having a formal diagnosis of developmental language disorder (DLD), and (3) the parent having access to a digital device (computer, tablet PC, mobile phone) and an internet connection to engage with intervention materials to support their child, (4) parents who are fluent in English.

The participants comprised ten parents (eight from the UK, one from Australia, and one from Ireland) of children with DLD. All the parents who voluntarily participated in this study were female and identified as white. Eight of them reported their relationship status as ‘married’, one as ‘single’, and one as ‘separated’. The educational status of the participants was recorded as follows: five with high school graduation (HSG), four with a bachelor’s degree (BA), and one with a master’s degree (MA). They reported their employment status as six part-time working (PT), one full-time (FT), one self-employed (SE), and two unemployed (UE). The fathers’ education and working (E&W) status is also presented in Table 1. The mean age of the parents was 40.4 years, and that of the children with DLD was 7.8 years. Two fathers and one mother were recorded as having DLD themselves, and three fathers had no DLD but were diagnosed with another neurodevelopmental disorder.

Parent C withdrew from the study after three weeks of enrolment, so their data are missing from the analysis (Table 1). However, we have maintained the original pseudonyms for consistency and clarity in the study documentation.

### 2.2. Measures

#### 2.2.1. The Parents’ Beliefs about Children’s Emotions (PBACE) Questionnaire 

PBACE, which was developed by Halberstadt et al. [16] to determine the validity of “seven different beliefs that parents have about their children’s emotions” (p. 12), was used in this study to understand whether the proposed intervention program makes any change in parents’ beliefs about their child’s emotions. PBACE uses a Likert scale with responses ranging from 1 to 6, from “strongly disagree” (indicated as 1) to “strongly agree” (indicated as 6). It comprises 33 items across seven reliable subscales, each demonstrating good internal consistency as measured by McDonald’s omega [17]. ‘Autonomy’ (7 items) and ‘Control’ (5 items) subscales were used weekly. The full scales of PBACE were also completed by participants pre- and post-test, including seven subscales: ‘Cost of Positivity’, ‘Value of Anger’, ‘Manipulation’, ‘Control’, ‘Parental Knowledge’, ‘Autonomy’, and ‘Stability’. 

Examples from the ‘Autonomy’ subtest are “When children are sad, they need to find their own ways to move on” and “It’s usually best to let a child work through being sad on their own”, and from the ‘Control’ subtest are “Children can control how they express their feelings” and “When children are very angry, they can control what they show to others” ([16], p. 20). Some of the ‘Cost of Positivity’ items include “When children are too happy, they can get out of control” and “Children who feel emotions strongly are likely to face a lot of trouble in life”. Examples of ‘Manipulation’ items are “Children use emotions to manipulate others” and “Children sometimes act sad just to get attention”. Some of the ‘Stability’ items are “Children’s emotions tend to be long-lasting” and “Children’s emotional styles tend to stay the same over time”. Thus, in our study, if scores decrease for these subscales, it indicates that parents’ beliefs have changed and that children should be supported and guided in understanding their emotions. 

Example items from the ‘Value of Anger’ subscale include “It is useful for children to feel angry sometimes” and “Children’s anger can be a relief to them, like a storm that clears the air”. Lastly, example items from the ‘Parental Knowledge’ subscale are “Parents should encourage their child to tell them everything they are feeling” and “It is important for children to tell their parents everything that they are feeling”. Therefore, increasing scores for the ‘Value of Anger’, and ‘Parental Knowledge’ subscales could signify a more open and accepting attitude regarding expressing emotions like anger and a stronger emphasis on the importance of transparent communication between parents and children.

#### 2.2.2. Weekly Interview (WI) and End Interview (EI)

Regular weekly interviews (WI) were conducted to monitor participants’ progress, answer their questions, and resolve any issues at the end of each week, as described in the study by Lacava et al. [18]. The first author arranged phone call schedules with participants at their convenience every week. This allowed the researchers to ensure that participants understood the intervention program. Participants were asked basic questions each week to gather feedback on the intervention. Additionally, more detailed information was collected from parents through video call end interviews (EI). Seven parents agreed to be interviewed at the end of the intervention and were given the written transcripts to review their statements and make any changes if they were dissatisfied. They did not change anything after the revision, but some wanted to add more details to the transcriptions to make them more transparent.

### 2.3. Intervention Materials

Intervention materials were prepared using Articulate 360 Software (https://www.articulate.com/360/, accessed on 25 July 2024), then downloaded as a SCORM package and uploaded to the University of Bath’s Moodle page. Each participant was provided with personal sign-in details to access the activities. The intervention materials consisted of three phases: Phase I included activities on basic emotions, and Phase II included activities on complex emotions to support children with DLD through the guidance of their parents. The impact of these phases was presented in another research paper [19], and results indicated that such activities encouraged active participation of both children and parents, fostering a collaborative environment that facilitated the practical application of emotion recognition skills. There was an improvement in the children’s emotion recognition abilities as well as social–emotional outcomes. 

Phase III comprised interactive activities exclusively for parents, incorporating interactive information and practices from the latest studies. These activities aimed to enhance parents’ knowledge of the importance of emotional recognition (ER) abilities for children with DLD, enabling parents to guide their children in daily routines. Phase III materials were chosen to enhance parents’ awareness of ER and were prepared in an interactive format. The interactive activities for Phase III were prepared using materials that have proven effective in previous studies. ‘The Feeling Wheel’ [20] was used to extend the use of emotional words, and ‘the Feeling Iceberg’ [21] was used to broaden the language of emotions and increase understanding of what is happening beneath the emotions. Additionally, some strategies for parents to support the ER abilities of children were taken from Siegel and Bryson’s [9] book. We received permission from the authors to use the pages included in our intervention program. 

Parents were flexible in engaging in any phases and activities related to their unique family situation. We could track participants’ progress during the intervention and we observed that six out of ten parents completed all phases. Phase III was the main part of improving parents’ beliefs about their children’s emotions, but Phase I and Phase II (which mainly include activities for children) would also have indirectly impacted parents’ awareness.

### 2.4. Procedure

To recruit participants, the aims and purpose of the study were advertised in the ’Engage with Developmental Language Disorder (E-DLD)’ project’s monthly research emails [22], and an informative leaflet was shared on social media platforms. Parents interested in participating in the study contacted the first author via email. Participants were provided with a detailed information sheet and debriefing sheet about the research, along with consent and assent forms. After completing these forms, they were invited to complete the participant characteristics questionnaire.

In this study, the parent-led online ER intervention is the independent variable, while the parents’ beliefs about their children’s emotion recognition abilities are the dependent variable. The intervention aims to impact and measure changes in the parents’ beliefs over time. This relationship is evaluated through various intervention phases, including baseline and follow-up assessments, and is supported by qualitative data from weekly and end interviews with the parents.

The pretest assessment involved completing a participant characteristics questionnaire and being randomly assigned to one of three baseline schedules to gather initial data on parents’ beliefs about their children’s emotional regulation (ER) abilities. The baseline phase includes a record of the continued beliefs of parents without the intervention, establishing a control period where weekly surveys track any changes. The intervention phase comprised three phases of activities designed using Articulate 360 Software and uploaded to the University of Bath’s Moodle page. Phase I focused on basic emotions, Phase II on complex emotions, and Phase III on enhancing parents’ knowledge of ER abilities. Participants were randomly assigned to one of three baseline schedules to commence the intervention program. In the first group, no one commenced Phase III—focused on enhancing beliefs regarding the importance of children’s emotions—before the 3rd week. In the second group, no one began Phase III before the 4th week; in the third group, no one initiated the intervention before the 5th week. Although Phase III is considered a follow-up phase, participants continued to use it actively, so it was included in the intervention phase for IRD calculations. Weekly interviews were conducted throughout the intervention to monitor progress and gather feedback. Participants were also asked to provide the most suitable times to receive a weekly audio or video call (depending on their preference). During these calls, participants were asked if they had any questions or feedback about the resources and materials; these calls lasted approximately 5–10 min per participant. At the end of their 13-week involvement in the program, participants were also interviewed to understand their thoughts about the intervention more deeply. This comprehensive approach ensured detailed tracking of changes in parents’ beliefs and allowed for ongoing adjustments to the intervention program. 

QuestionPro Software (https://www.questionpro.com/, accessed on 25 July 2024) was used to collect demographic information and administer the weekly questionnaires. Interviews were conducted with participants via Microsoft Teams (https://www.microsoft.com/en-gb/microsoft-teams/group-chat-software/, accessed on 25 July 2024) meetings, and the interviews were recorded only after obtaining the participants’ permission at the beginning. After the interviews, participants had the opportunity to read the transcribed interviews and make any changes if they were dissatisfied with the transcription. 

To ensure the reliability of our study, we implemented several measures to maintain high treatment integrity throughout the intervention. Detailed guidelines and instructional materials, including step-by-step instructions, were provided to participants to ensure clarity and consistency. Regular weekly fidelity check-ins via phone or video call allowed researchers to monitor adherence to the intervention protocol, address any challenges, and provide additional support. Participants’ engagement with the intervention was tracked through weekly logs detailing which phase the participant was engaging in, and the duration and frequency of the use of activities were regularly reviewed by researchers. An open feedback mechanism was also established, enabling participants to provide continuous input, allowing for real-time adjustments and improvements. These measures ensured that the intervention was delivered as intended, enhancing our findings’ reliability and validity.

A PhD researcher with experience in similar processes in children with DLD checked the data and IRD calculations for all participants independently. Additionally, another researcher conducted a qualitative thematic analysis of the interview transcripts. This researcher independently coded the data and created initial themes. Subsequently, both researchers engaged in a discussion to resolve discrepancies and refine the themes.

Social validity was assessed by conducting a pilot study before this intervention with two volunteer parents and one clinical psychologist. This allowed us to modify the intervention accordingly. Feedback from participants during weekly interviews and end-of-study evaluations further ensured that the intervention was perceived as valuable and relevant by the parents. These measures enhance the reliability and validity of our findings and provide a comprehensive understanding of the intervention’s impact.

The ethics of this study were reviewed and approved by the University of Bath Ethics Committee (REF: 22 139).

### 2.5. Design and Analysis

Parker et al.’s [23] improvement rate difference (IRD) for single-case research was utilized to analyze weekly data. The IRD enables the quantitative assessment of intervention impact within the same individuals by calculating the control (baseline phase in this study) and intervention phases. In this study, the IRD is calculated by comparing the proportion of improved data points in the baseline phase with the proportion in the intervention phase. Reduction in data score from baseline to intervention indicates improvement. In order to interpret the IRD values, an overlap between the baseline and intervention was identified. Improvement rate difference is calculated for baseline and intervention phases after removing overlapping data. Lastly, the baseline success rate is subtracted from the intervention rate to calculate the IRD value. Vannest and Ninci [24] stated that IRD is “small or to have a questionable effect at 0.50 and below, moderate from 0.50 to 0.70, and large and very large at 0.70 and 0.75, respectively”. They also proposed that the effect size should be considered in relation to three key factors: the objectives or requirements of the individual, the nature of the intervention, and setting. In order to understand the individual IRD values fully, individual WIs and EIs were used. 

Semi-structured weekly interviews (WI) and end interviews (EI) were also analyzed by means of thematic analysis to see changes in the parents’ beliefs across participants. The six phases of Braun and Clarke’s [25] thematic analysis approach were followed. Firstly, immersion in the data was conducted to gain a deep understanding. Then, initial codes were generated to label key concepts. Next, themes were identified by looking for recurring patterns. Potential themes were reviewed to ensure accuracy and relevance. Subsequently, themes were defined and named to provide clear descriptors. Finally, findings were synthesized into a comprehensive report, presenting themes with illustrative quotes.

## 3. Results

This section will present paired sample *t*-test results, nonconcurrent multiple baseline data, and semi-structured interview results. Paired sample *t*-test results of the full PBACE will be presented for all participants. Seven parents consistently completed the ‘Control’ and ‘Autonomy’ subtests of PBACE every week, while three others have some missing data for some weeks (Figure 1 and Figure 2). WIs were used to receive feedback from parents weekly through phone calls, and EIs were completed with seven participants at the end of the intervention.

### 3.1. Paired Sample t-Test Results

Parents completed the full PBACE scale to determine whether the intervention changed their beliefs. The results showed a significant difference in the scores for Manipulation pre-intervention (M = 12.9, SD = 3.51) and post-intervention (M = 6.5, SD = 1.35); t(9) = 5.018, *p* = 0.001; Parental Knowledge pre-intervention (M = 7.3, SD = 2.54) and post-intervention (M = 12.2, SD = 2.78); t(9) = −6.391, *p* = 0.001; Stability pre-intervention (M = 15.4, SD = 2.06) and post-intervention (M = 11.1, SD = 1.52); t(9) = 5.095, *p* = 0.001; and Value of Anger pre-intervention (M = 16.4, SD = 1.42) and post-intervention (M = 24.0, SD = 3.43); t(9) = −5.919, *p* = 0.001. As shown in Table 2, four out of the seven subscales were statistically significant. Parents’ beliefs about the ‘Control’, ‘Autonomy’, and ‘Cost of Positivity’ subscales were not significant. 

To determine whether the intervention impacts parents’ beliefs, the ‘Control’ and ‘Autonomy’ subtests of PBACE were collected from parents weekly. These data will be interpreted along with WI and EI to enrich the analysis. 

### 3.2. Multiple Baseline Results

The IRD for each child’s Baseline (B) versus Intervention (I) shifts were calculated to determine the IRD for all participants (Table 3). Figure 1 and Figure 2 illustrate single-case data points for Control and Autonomy data, facilitating B versus I contrasts as recommended by Parker et al. [23]. However, studies have highlighted—and it is mentioned in the analysis section—that categorizing the IRD values as small, medium, or large might be insufficient [24]. Therefore, participants’ settings, interventions, and individual factors will be explained along with the IRD values for a comprehensive understanding. 

Parent D, Parent E, and Parent H’s beliefs about their child’s autonomy showed an IRD of 0.77, 0.85, and 0.75, respectively, indicating a large effect size; whereas their Control IRD was found to be 0 (no effect), 0.25 (small effect), and 0.5 (medium effect), respectively. Parent D and Parent E completed all phases, but their beliefs about children’s emotions showed a small effect in terms of ‘controlling emotions’. Parent D, a single mother working full-time, stated in weekly calls that “it is not easy to find some time to teach and assist DLD D, so sometimes I (Parent D) let DLD D engage in activities by himself” (EI). Although Parent D stated she hardly finds time to help her child, there is still a large change in her autonomy beliefs. The other differences between these participants are that Parent D and Parent E completed all phases, Parent H only guided and helped her child to complete Phase I, but did not engage with other phases.

Parent A’s beliefs for DLD A show that both Control and Autonomy IRD were 0.66, indicating a moderate effect size. Parent B also showed a moderate change in their autonomy belief with an IRD value of 0.66, and also Parent I’s control belief showed moderate effect with 0.63. Parent B’s IRD value was 0.42 for Control data, while Parent I’s Autonomy IRD was found to be 0.23, indicating a small effect size. While IRD values show that Parent I’s beliefs about her child’s control of emotions had a moderate effect and autonomy of emotions had a small effect, interview data provide more detail indicating that this small effect might have yielded an important contribution for parents, as she stated, “It would be true to say that I had underestimated her and am now doing my best to appreciate her incredible qualities and even her thought process. I’m sure she is smarter, fitter, faster than I think” (EI). Parent B’s interview data also supported this effect as she stated activities increased her awareness and use of emotion-aware communication. The only difference is that while Parent B completed all phases, Parent A only assisted her child in engaging with activities in Phase I as she thinks DLD A needs more support to learn basic emotions first. Although they engaged with different phases, the intervention had a moderate impact on both Parent A’s and Parent B’s autonomy beliefs about their child’s emotions. These effects were also supported by the interview data as both parents stated that intervention enhanced their awareness to identify what to support and that engaging with their child with various activities was fun. 

Parent F, Parent G, Parent J’s data showed a small effect size for both Control data and Autonomy data. The IRD value of Control data was 0.03, and Autonomy data was 0.42 for Parent F; the IRD value of Autonomy and Control data was 0.33 for Parent G; and Parent J’s IRD was 0.13 for Autonomy and 0.03 for Control data, indicating a small effect size. So, the Control and Autonomy data of these participants showed a small effect. However, it is important to consider that Parent F and Parent G completed all phases, while Parent J completed only Phase I. Additionally, Parent K’s data show that she completed Phase I and Phase II. The intervention had a small effect on her control beliefs about her child, with an IRD value of 0.25, and a moderate effect on her autonomy beliefs, with an IRD of 0.5. 

As suggested by Parker et al. [23], all participants’ IRD values can be found by calculating the average of all baseline and intervention scores for participants. All participants’ IRD was 0.27 for Control data and 0.52 for Autonomy data. Thus, the intervention appears to positively impact parents’ beliefs about autonomy, with the effect size leaning towards moderate. For the Control data, results also show improvement with the intervention, but the effect size is small. When drawing conclusions, it is crucial to consider individual differences, the variability of the data, and the practical significance of these differences in effect sizes.

### 3.3. Interview Results

In this section, weekly and end interview data will be presented to understand the change in parents’ beliefs about their child’s emotion throughout the intervention process in a deeper way. All parents were contacted weekly through phone calls to conduct WI, but seven of them also agreed to be interviewed at the end of the intervention. Parent D, Parent E, and Parent H were not available for an EI. Interview data are presented under three themes (Figure 3).

#### 3.3.1. Parents’ Awareness of the Importance of ER for Children with DLD

Interview data indicated that the intervention helped parents to understand their child’s unique experience of emotions. Parent K expressed that intervention enhanced her awareness of the need for clearer explanations of emotions: “It’s made me more conscious of explaining things easier and perhaps explaining what those words mean […]. That might mean something else” (Parent K). Parent I also highlighted that “I [Parent I] learned from my child that most emotions can be spoken about in simple terms when one reduces them down to their core”. Additionally, Parent K’s acknowledgement of the interplay between language and emotion recognition illustrates a significant shift in perception, as stated by Parent K: “I hadn’t even thought about reading people’s emotions as being something that that a person would struggle with. I just thought it was to do with words and reading and stuff. I didn’t think it would affect how someone interprets body language. And so yeah, that was good for me” (Parent K). 

Parent F also underlined the value of visual activities and their impact on facilitating discussions about complex emotions: “I in the past found that difficult to explain to DLD F what those things [complex emotions] mean. The resources, the way you explained them, and the charts to show, like the faces or the body, just made it. It presented sort of more of a visualization of those complex emotions which made it a bit easier to explain” (Parent F). This demonstrates how the intervention enhanced their ability to explain such concepts to their child. 

Parent A emphasized improved identification of emotions in their child and increased readiness to address emotional needs: “I think it’s been great for me as a parent cause I’ve now been able to identify maybe feelings and emotions that he maybe doesn’t recognize as quickly […]. It helped me as a parent to identify any areas that we may need to talk about going forward or I may need to support him in recognizing and processing particular emotion”. (Parent A). These quotes indicate the significant impact on parental awareness of emotion recognition for their children with DLD, which helps them in guiding and supporting effective ER communication.

#### 3.3.2. Emotion-Focused Communication and Engagement with the Child

The intervention facilitated more open and expressive dialogue between parents and their children regarding emotions. Parent J’s expression highlights a shift towards proactive emotion labeling within the family, fostering emotional awareness and discussion. “I think I label my emotions more now like I say. Ohh, mummy’s feeling frustrated. You know, mummy’s feeling sad, and my other, my other children, they have also started doing it as well. So, we talk a little bit more about emotion and how we’re feeling” (Parent J).

Similarly, Parent I emphasizes the importance of the use of emotional language during times of stress: “It was nice to have the vocab for more complex emotions available and yes, we were starting to make greater use of them, but in time of stress, it was the simpler ones which gave my child the freedom to speak of their feelings with confidence at this time” (Parent I).

Parent B and Parent G underlined the value of indirect approaches to discussing emotions, enabling the child to engage without feeling pressured. Parent G described the enjoyable interactions sparked by discussions about facial expressions, demonstrating how the intervention fostered playful engagement and deeper understanding: “[…] it was useful cause I was able to bring up emotions without putting the focus on him […]. It was useful for me to not feel like I’m putting pressure on him. So, I was able to sort of like point and then show him point to the hand, your hands might get clammy. I’d point to the forehead your forehead might start to swear. And a couple of times where something’s happened to him. He’s sort of got the body thing and he sort of showed me like, ‘My legs went to Jelly when he was really nervous about race’ and that sort of thing, so it is definitely helped bring in emotions to live in the body” (Parent B).

“What DLD G and I [Parent G] thought of the facial expressions. She found some of them quite funny and and there was sometimes a discussion. Ohh she sounds very jealous. It was quite fun sometimes to engage in DLD G in talking about the cases” (Parent G).

Interview data showed that the intervention materials and activities facilitated richer, more nuanced conversations about emotions within the parent–child dynamic, which promoted the use of more emotional literacy and connection.

#### 3.3.3. Integration of Emotions into Daily Life in a Natural Way

Data showed that parents started to initiate discussions about emotions in their everyday interactions with their children. Parent B’s statement highlights a shift from solely addressing situations to openly discussing emotions and normalizing feelings for both her child and herself in that she stated that “I’m able to rather than just talk about situation, I can then talk about my feelings with my child or my boss and how it’s OK to be nervous” (Parent B). Parent J also stated that she labels and uses emotional words naturally: “He [DLD J] started to label his emotions a little bit more and because it was something that I tried to do every day throughout the day, so he started feeling a little bit more relaxed about talking about emotions”. Parent K also underlined that the intervention made her more conscious about explaining emotional words: “I don’t think he understands some words a lot of the time. So, it’s made me more conscious of explaining things more easily and perhaps explaining what those words mean. That might mean something else”.

Data also showed that the intervention process increased parents’ awareness about how to approach and talk about emotions with their children by using activities and natural conversations within daily life, rather than structured and personal conversations. Parent I stated that “it has been an eye opener for me to really study how she dislikes discussing emotions, especially negative ones. She tries to change the topic rather than get into a nitty-gritty discussion of how she or others are feeling, etc. But if I make it into an abstract game, we make much more progress, and she likes to act out roles or do rough drawings rather than make it personal”.

These expressions indicate that parents shifted their daily communication and interaction, moving beyond problem-solving to nurturing emotional intelligence and resilience in their children. The intervention empowered parents to understand the importance of emotional awareness and the use of strategies in their daily routines.

## 4. Discussion

### 4.1. Shifts in Parental Communication Patterns: ER Matters

The intervention significantly affected parents’ awareness about their child with DLD ER abilities. Parents reported enhanced awareness and a deeper understanding of their child’s emotional experiences through the activities and discussions. Parallel findings were also reported by Halberstadt et al. [16], who highlighted that those parents with a strong belief in the importance of autonomy—that is, the belief that children should independently manage their negative emotions—tend to be less supportive and more unsupportive in response to their children’s negative emotional expressions. Rangel-Rodriguez and Blanch [26] also found that emotion-focused interventions can improve parental awareness of the importance of using emotion-related conversations with children with complex communication needs. This underlines the importance of interventions designed to enhance parents’ awareness of the importance of ER, specifically addressing beliefs around autonomy in emotional responses. This study differs from other studies in that it was designed with various online activities for the use of parents in a flexible way as an intervention method for children with DLD. The intervention facilitated positive shifts in parental communication patterns, encouraging more emotion-aware dialogue between the parent and the child.

Parents reported feeling more comfortable discussing emotions with their children and observed increased willingness among their children to engage in such conversations. Parents reported incorporating discussions about emotions into various contexts, such as mealtimes, bedtime routines, and everyday activities. This emotion-focused communication reflects a deeper understanding of emotions’ roles in children’s socio-emotional development. Similar findings were reported by Beck [27], who emphasized the importance of using emotional learning in everyday experiences to promote holistic child development. In this study, providing activities for parents to facilitate conversations and discussions about emotions during the daily routine promoted the use of emotional literacy both for parents and children.

### 4.2. Individual Variations in Intervention Impact

The intervention, designed to enhance emotional understanding among children with DLD and their parents, led to significant changes in parental attitudes towards children’s emotions. After the intervention, there was an observable increase in scores for the ‘Value of Anger’ and ‘Parental Knowledge’ subscales, suggesting that parents became more accepting of anger and emphasized the importance of transparent emotional communication. Scores for the ‘Cost of Positivity’, ‘Manipulation’, and ‘Stability’ subscales indicated a shift in parental beliefs towards supporting and guiding children in understanding their emotions rather than viewing strong emotions as problematic or manipulative. These changes reflect the impact of the intervention, particularly Phase III, which focused on interactive activities for parents to enhance their knowledge and application of emotional regulation strategies, as well as the indirect influences from child-centered activities in Phases I and II.

In this study, while the pre- and post-tests of the ‘Autonomy’ and ‘Control’ subtests did not yield statistically significant results, using IRD values alongside within-individual WI and between-individuals EI comparisons revealed improvements that were not apparent from the statistical assessments. This discrepancy highlights the importance of integrating individual-level quantitative data with qualitative insights from interviews, suggesting a marginally significant enhancement in autonomy among participants. These findings suggest the value of employing mixed methods to fully capture complexities and interventions’ differential impacts on individual participants.

The impact of the intervention varied among participants’ parents. While some parents experienced significant shifts in their beliefs and communication patterns, others reported more subtle changes. This variation may be related to factors such as the level of engagement with the intervention materials, cultural and geographical differences, and parents’ relationships, working, and education statuses. For instance, Parent K’s level of engagement had been affected by the family health situation; they neither engaged with activities nor filled out the questionnaires, as can be seen in Figure 1 and Figure 2. This might be the reason for the subtle change in their beliefs. Additionally, parents’ own experiences with emotional expressions and interpersonal relationships may impact their expectations and interpretations of their child’s ER skills. Moreover, because parents of children with developmental disabilities experience more stress compared to those of typically developing children [28], providing flexible and suitable support could ease parents’ lives. By recognizing and understanding these individual differences, professionals can better support parents with suitable materials [29].

The variability in the impact of the intervention among participants underlines the influence of individual and contextual factors on the outcomes. Significant shifts in beliefs and communication patterns were observed in parents like Parent D, Parent E, and Parent H’s autonomy beliefs, demonstrating large effect sizes in their IRD values. Conversely, Parent J’s beliefs were characterized by small effect sizes, possibly due to lower engagement with activities as she did not engage with Phases II and III. As we stated before, parents and children had the flexibility in this study to engage with any phases and activities regarding their child’s needs and interests, so personal circumstances also impacted the depth of intervention interaction. These examples highlight the necessity of flexible interventions to accommodate the diverse backgrounds and needs of families to optimize the benefits of ER interventions for children with DLD and their parents.

### 4.3. Implications for Practice and Future Research

The implications of this study can be presented as two-fold. Practically, the study indicated that all participating parents showed some level of improvement in their beliefs and support regarding their child’s emotional skills. Thus, parent-led interventions that are flexible, adaptable, and deeply embedded in the daily routines of families could more effectively assist families of children with developmental language disorder (DLD) and their varied needs.

Academically, the study revealed that the more actively parents participate in and learn from online activities, the more positively their attitudes shift towards supporting their children’s ER skills. Based on these findings, future research could investigate the link between parental engagement and interaction styles, and their awareness and support for their children’s ER abilities.

It is also important to note that quantitative results in this study showed that online, parent-led intervention did not affect parents’ control, autonomy, and cost of positivity beliefs. Adding more activities to support parents regarding these subjects might be useful.

### 4.4. Strengths, Limitations, and Future Directions

The present study highlights the significant impact of a parent-led ER intervention on parental beliefs and communication patterns in families of children with DLD. The study’s most significant strength is the richness of the data and its interpretation of these data in both qualitative and quantitative ways. 

In this study, all participants were mothers. This outcome was not intentional as our recruitment strategy aimed to involve all parents, regardless of gender. The study invitation was disseminated broadly via social media platforms and a database of parents with children with DLD without specifying gender. However, only mothers chose to participate. While this reflects a gender imbalance in participation, it does not imply that fathers were excluded or could not participate. This finding highlights an area for further exploration regarding the barriers to participation for fathers in such interventions. Future research could investigate strategies to encourage more diverse parental involvement, ensuring that mothers and fathers can equally contribute to and benefit from parent-led interventions.

Although conducting a case study is not a limitation but rather a strength to provide deeper understanding, future studies could include a larger sample to ensure the generalizability of findings across different populations. Additionally, although 13 weeks of assessment provides valuable and consistent information about the change in parents’ beliefs, longer-term follow-up assessments could provide valuable insights into the sustainability of intervention effects over time. By empowering parents with related activities and materials, the intervention can promote open dialogue about emotions and integrate emotional awareness into daily life.

## 5. Conclusions

The primary aim of this study was to explore the impact of a parent-led online intervention on parents’ beliefs regarding their children’s emotional understanding and regulation, specifically targeting children with DLD. The intervention examined how parents’ perceptions of their children’s ER skills might change through their engagement with designed online activities to enhance emotional awareness and communication. A nonconcurrent multiple baseline design facilitated a detailed examination of changes in parental beliefs about their children’s emotions over time. The intervention influenced parental beliefs, leading to a shift towards understanding the importance of guiding and supporting their children’s ER skills. This change was seen in improvements in parents’ beliefs, as measured by the full scale of the PBACE Questionnaire. Individual IRD calculations and qualitative data also supported this as quotes show a shift towards more supportive beliefs about the need to guide and support their children’s ER skills. Qualitative data from interviews highlighted the themes of increased parental awareness and the integration of emotion-focused communication and engagement practices in daily interactions with their children.

## Figures and Tables

**Figure 1 healthcare-12-01571-f001:**
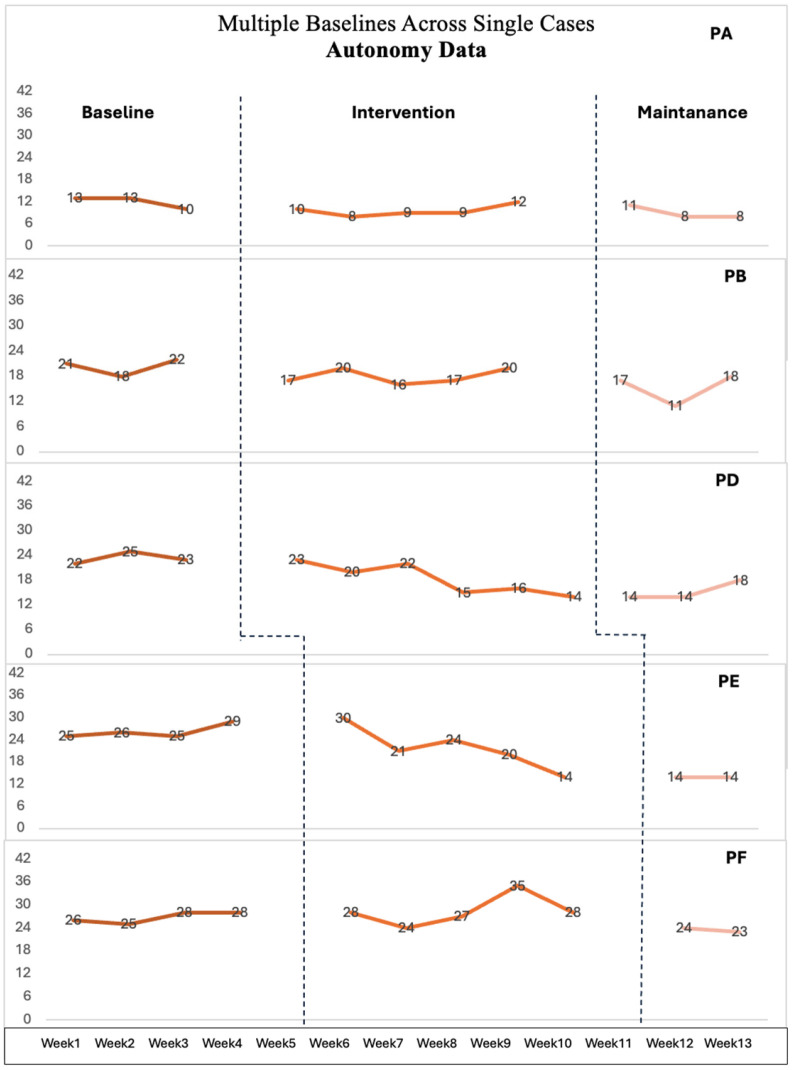
The scores of weekly Autonomy data (3-4-5 weeks baseline).

**Figure 2 healthcare-12-01571-f002:**
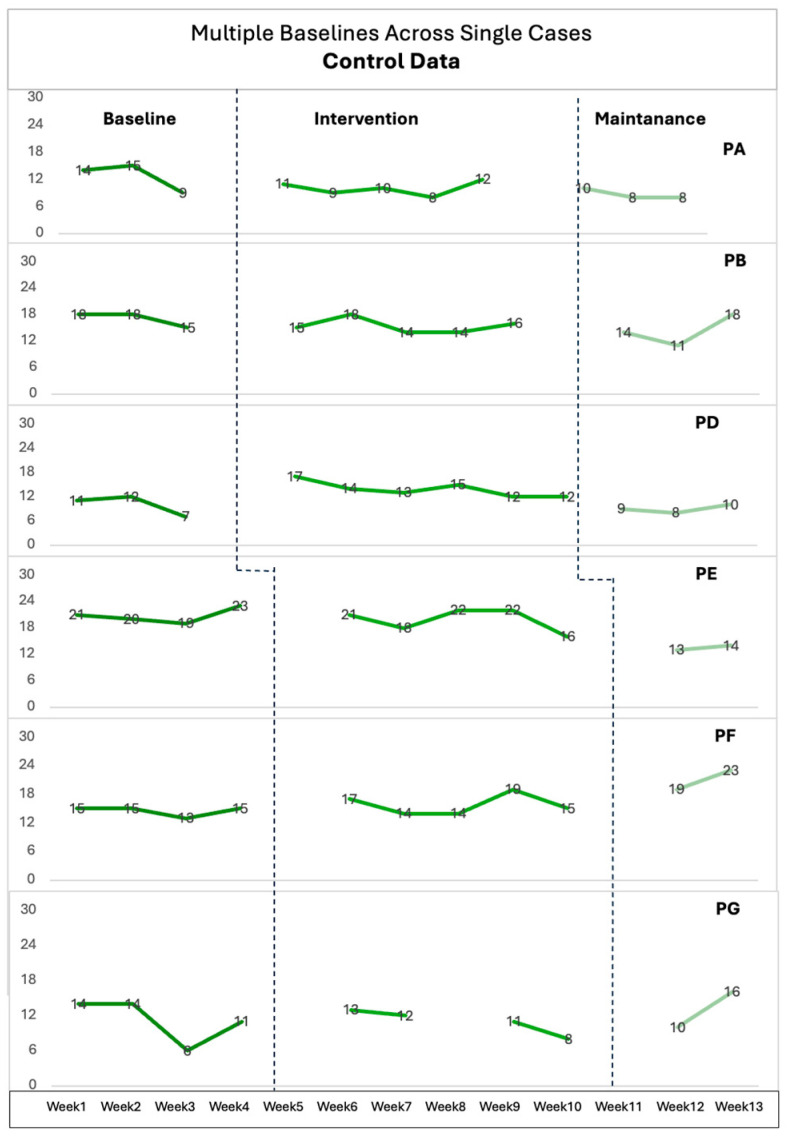
The scores of weekly Control data (3-4-5 weeks baseline).

**Figure 3 healthcare-12-01571-f003:**
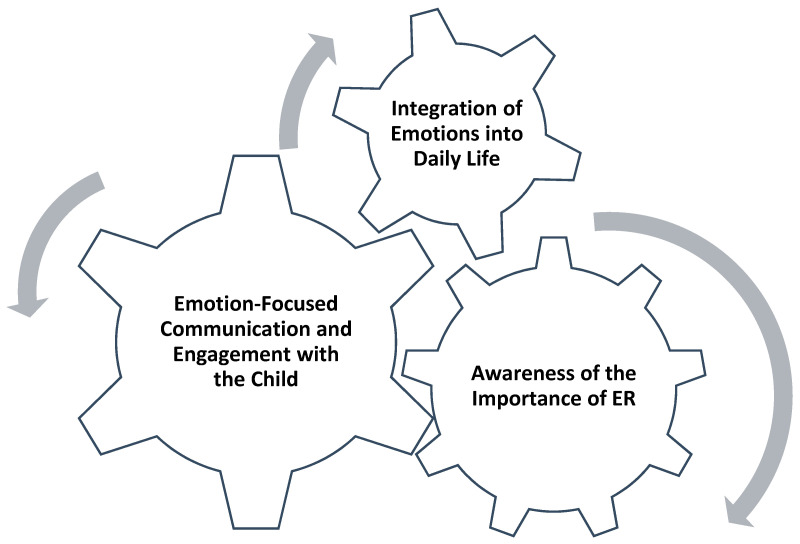
Themes showing the transformation of parents’ beliefs.

**Table 1 healthcare-12-01571-t001:** Participants’ demographics.

Participants	Parent’s Age	Child’s Age	Parent’s Gender	Child’s Gender	DLD Heredity	Relationship Status	Ethnicity	Country	E&W, Mother	E&W, Father
Parent A	36	6	F	M	Father	Married	White	UK	HSG-UE	VT-FT
Parent B	35	6	F	M	Father	Married	White	UK	BA-PT	BA-FT
Parent D	33	7	F	M	No	Single	White	UK	HSG-FT	VT-FT
Parent E	44	11	F	F	Father *	Married	White	UK	BA-PT	BA-SE
Parent F	40	7	F	F	Father *	Separated	White	UK	HSG-PT	HSG-FT
Parent G	40	9	F	F	No	Married	White	UK	BA-PT	MD-FT
Parent H	48	6	F	M	No	Married	White	Australia	BA-PT	BA-FT
Parent I	47	9	F	F	Mother	Married	White	Ireland	MD-SE	BA-SE
Parent J	40	7	F	M	No	Married	White	UK	HSG-UE	HSG-FT
Parent K	41	10	F	M	Father *	Married	White	UK	HSG-PT	HSG-FT
Total	40.4	7.8	10 F	6 M, 4 F	6/10 Heredity	8 Married, 1 Single, 1 Separated	10 White	8 UK, 1 AU, 1 Ireland	5 HSG, 4 BA, 1 MD; 6 PT, 1 FT, 1 SE, 2 UE	4 BA, 3 HSG, 2 VT, 1 MD; 8 FT, 2 SE

E&W = education and working status: PT = part time, FT = full time, UE = unemployed, VT = vocational training, BA = bachelor’s degree, HSG = high school graduate, SE = self-employed, MD = master’s degree, M: male, F: female. * = Parent has no DLD diagnosis but has a diagnosis of other neuropsychological disorder.

**Table 2 healthcare-12-01571-t002:** Paired sample *t*-test for the full PBACE.

Pre-Test, Post-Test Pairs	Mean	Standard Deviation	Standard Error Mean	Lower	Upper	t	df	Two-Sided *p*
Autonomy	3.50	4.95	1.56	−0.04084	7.04084	2.236	9	0.052
Control	−2.30	6.36	2.01	−0.6.8500	2.25000	−1.144	9	0.282
Cost of Positivity	−0.30	3.40	1.07	−2.73291	2.13291	−0.279	9	0.787
Manipulation	6.40	4.03	1.27	3.51483	9.28517	5.018	9	0.001
Parental Knowledge	−4.90	2.42	0.76	−6.63432	−3.16568	6.391	9	0.001
Stability	4.30	2.66	0.84	2.39089	6.20911	5.095	9	0.001
Value of Anger	−7.60	4.06	1.28	−10.50481	−4.69519	−5.919	9	0.001

**Table 3 healthcare-12-01571-t003:** Analysis of multiple baseline scores of PBACE.

		B versus I Proportions	IRD	95% CI	Completed Phases
DLD A	Control	1/2, 8/0	0.66	[0.13, 1.2]	I
	Autonomy	1/2, 8/0	0.66	[0.13, 1.2]	
DLD B	Control	1/2, 6/2	0.42	[−0.2, 1.03]	I, II, III
	Autonomy	1/2, 8/0	0.66	[0.13, 1.2]	
DLD D	Control	1/2, 3/7	0	[−0.62, 0.62]	I, II, III
	Autonomy	0/3, 7/2	0.77	[0.51, 1.05]	
DLD E	Control	3/1, 7/0	0.25	[−0.17, 0.67]	I, II, III
	Autonomy	0/4, 6/1	0.85	[0.6, 1.12]	
DLD F	Control	1/3, 2/5	0.03	[−0.5, 0.58]	I, II, III
	Autonomy	0/4, 3/4	0.42	[0.06, 0.8]	
DLD G	Control	2/2, 5/1	0.33	[−0.24, 0.91]	I, II, III
	Autonomy	2/2, 5/1	0.33	[0.24, 0.91]	
DLD H	Control	2/2, 5/0	0.5	[0.01, 0.99]	I
	Autonomy	1/3, 5/0	0.75	[0.33, 1.17]	
DLD I	Control	1/4, 5/2	0.63	[0.17, 1.09]	I, II, III
	Autonomy	3/2, 5/1	0.23	[0.29, 0.76]	
DLD J	Control	1/4, 5/2	0.03	[−0.43, 0.49]	I
	Autonomy	1/4, 2/4	0.13	[0.38, 0.65]	
DLD K	Control	3/1, 5/0	0.25	[−0.17, 0.67]	I, II
	Autonomy	2/2, 5/0	0.5	[0.01, 0.99]	
TOTAL	Control	19/20, 51/19	0.27	[0.087, 0.46]	
	Autonomy	11/28, 54/13	0.52	[0.35, 0.69]	

## Data Availability

Data are available from the authors upon request.

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
