# Peer review of "Enhancing Parental Understanding of Emotions in Children with Developmental Language Disorder: An Online Parent-Led Intervention Program"

_healthcare, 2024, doi:10.3390/healthcare12161571_

Round 1
Reviewer 1 Report
Comments and Suggestions for Authors
The paper explores the importance and mechanisms of parental support in their DLD children's development of perceiving and processing emotions, following a protocol of weekly assessments and self-aware interventions. There are some elements which could be improved, see attached file.

Author Response
|
Response to Reviewer1’s Comments |
||
|
Thank you very much for taking the time to review this manuscript. We have thoroughly reviewed and revised the manuscript for grammatical accuracy and clarity, correcting verb tenses, improving sentence structures, and ensuring the appropriate use of vocabulary. Specific changes include addressing errors, ensuring proper subject-verb agreement, correcting spelling and typographical errors throughout the text. All these changes are shared in the re-submitted files using track changes for your review. |
||
|
Point-by-point response to Comments and Suggestions: |
||
|
Comments 1: Express the themes more clearly Response1: Themes are modified in the abstract (Line 21-23, p.1)
Comments2: Suggestion of DLD to be spelling out once more in the introduction Response1: The term “Developmental Language Disorder” is expressed once more at the beginning of the Introduction (Line 29, p.1)
Comments3: Some words should be modified in the text. Response3: The necessary changes are done throughout the manuscript. For example :(Line 45-46-64-96 etc.)
Comments4: Sentence needs a main verb Response4: Sentence is rewritten (Line 88-89, p.2).
Comments5: Format the paragraph as block Response5: WI and EI section is formatted (Line 163-173).
Comments6: Unify the style Response6: The style is unified and necessary changes are done throughout the manuscript for titles, figures and tables.
Comments7: You do not have ParentC. In that intentional? Response7: Participant C wanted to withdraw from the study after week three. We have now added this information into the manuscript (Line 120-123, ‘participants’ section).
Comments8: Supplementary materials expression needs editing. Response8: We shared the supplementary materials within the submission system, already. So, we have removed this expression. |
||
|
Additional clarifications |
||
|
We have added an acknowledgement section. We believe it will be important to acknowledge our colleagues who shared their expertise with us (Line between 623-625). We are open to any additional suggestions you might have. Thank you once again for your constructive feedback. |

Reviewer 2 Report
Comments and Suggestions for Authors
Dear researcher,
I appreciate your excellent efforts in this study. However, there are significant methodological concerns that need to be addressed.
(There is a fundamental issue in the study's methodology where researchers have not clearly delineated the relationship between the independent and dependent variables. This lack of clarity hinders other researchers interested in the subject from replicating the intervention. Moreover, key study procedures such as pretest assessment, baseline phase, intervention phase, and follow-up phase are ambiguously defined. Additionally, essential elements like treatment integrity, inter-rater agreement, and social validity were not addressed).
Despite clear efforts in the written content, the researchers did not succeed in implementing their study in a methodologically sound manner.
Author Response
|
Response to Reviewer2’s Comments |
||
|
Thank you very much for your constructive feedback on our manuscript. We have made significant revisions to address the methodological concerns you highlighted. Specifically, we have clearly delineated the relationship between the independent and dependent variables to facilitate replication of the intervention by other researchers. We have also provided detailed definitions and descriptions for key study procedures, including the pretest assessment, baseline phase, intervention phase, and follow-up phase. Additionally, we have addressed essential elements such as treatment integrity, inter-rater agreement, and social validity. All these revisions and clarifications are shared in the re-submitted files using track changes for your review. Point-by-point response to Comments and Suggestions: |
||
|
Comments 1: Defining the relationship between the independent and dependent variables. Response1: Definition of dependent and independent variable is explained as: “In this study, the parent-led online ER intervention is the independent variable, while the parents' beliefs about their children's emotional recognition abilities are the dependent variable. The intervention aims to impact and measure changes in the parents' beliefs over time. This relationship is evaluated through various intervention phases, including baseline and follow-up assessments, and is supported by qualitative data from weekly and end interviews with the parents.” (Line between 212-217)
Comments2: Key study procedures such as pretest assessment, baseline phase, intervention phase, and follow-up phase are needs to be defined. Response2: All phases are explained and defined under the “Procedure” section (Line between 218-240) as:
“The pretest assessment involved completing a participant characteristics questionnaire and being randomly assigned to one of three baseline schedules to gather initial data on parents' beliefs about their children's emotional regulation (ER) abilities. The baseline phase includes a record of the continued beliefs of parents without the intervention, establishing a control period where weekly surveys track any changes. The intervention phase comprised three phases of activities designed using Articulate 360 Software and uploaded to the University of Bath’s Moodle page. Phase I focused on basic emotions, Phase II on complex emotions, and Phase III on enhancing parents’ knowledge of ER abilities. Participants were randomly assigned to three baseline schedules to commence the intervention program. In the first group, no one commenced Phase III—focused on enhancing beliefs regarding the importance of children’s emo-tions—before the 3rd week. In the second group, no one began Phase III before the 4th week; in the third group, no one initiated the intervention before the 5th week. Although Phase III is considered a follow-up phase, participants continued to use it actively, so it was included in the intervention phase for IRD calculations. Weekly interviews were con-ducted throughout the intervention to monitor progress and gather feedback. Participants were also asked to provide the most suitable times to receive a weekly audio or video call (depending on their preference). During these calls, participants were asked if they had any questions or feedback about the resources and materials; these calls lasted ap-proximately 5-10 minutes per participant. At the end of their 13-week involvement in the program, participants were also interviewed to understand their thoughts about the intervention more deeply. This comprehensive approach ensured detailed tracking of changes in parents' beliefs and allowed for ongoing adjustments to the intervention program.”
Comments4: essential elements like treatment integrity, inter-rater agreement, and social validity were not addressed Response4: Detailed explanations about the elements like treatment integrity, interrater agreement and social validity outlined in the manuscript (Line 247-269)
|
||
|
Additional clarifications |
||
|
We are open to any additional suggestions you might have. Thank you once again for your constructive feedback. |

Reviewer 3 Report
Comments and Suggestions for Authors
Major issue
The results of this study show how important this type of intervention can be. It is sad, however, that the parents who agreed to participate are exclusively women. I think the authors should make a point of discussing this issue. This is not a ”parent-led intervention”, it is a ”mother-led intervention”.
Minor issues
1. Whatever hapenned to ParentC? They are missing everywhere.
2. The Material and Methods section needs a bit more care – it is rather ambiguously written, it lacks chronological order. For example, subsection 2.4 Intervention Materials should be placed before subsection 2.3 Procedure, that discusses the phases of the study, before those are even described.
Author Response
|
Response to Reviewer3’s Comments |
||
|
Thank you very much for your valuable feedback on our manuscript. We appreciate your recognition of the importance of our intervention and have made revisions to address your concerns. We have added a discussion on the gender imbalance among participating parents, emphasizing that while the study was indeed designed as a parent-led intervention, only mothers chose to participate. Additionally, regarding ParentC, they chose to withdraw from the study after three weeks of enrolment, which is why their data is missing. We have clarified this now in the manuscript and maintained the original pseudonyms of participants. We have also reorganized the Material and Methods section for better clarity and chronological order by placing subsection 2.4 Intervention Materials before subsection 2.3 Procedure. All these changes are highlighted in the re-submitted files using track changes for your review.
Point-by-point response to Comments and Suggestions: |
||
|
Comments 1: Using” mother-led intervention” rather than “parent-led intervention” Response1: We acknowledge the reviewer’s observation that the study appears to be mother-led rather than parent-led, as all participants were mothers. This was not the intention of our recruitment strategy, which aimed to involve all parents regardless of gender. We shared the study invitation broadly via social media platforms and a database of parents with children with DLD, without specifying gender. However, only mothers chose to participate in the study. While this reflects a gender imbalance in participation, it does not imply that fathers were excluded or could not participate. This finding highlights an area for further exploration regarding the barriers to participation for fathers in such interventions. Future research could investigate strategies to encourage more diverse parental involvement, ensuring that both mothers and fathers can equally contribute to and benefit from parent-led interventions. We have included a discussion on this gender imbalance and its implications in the revised manuscript to provide a comprehensive understanding of the study's context and to inform future research directions (Line between 586 to 595. Section 4.4 Strengths, Limitations and Future Directions).
Comments2: Clarifying what happened to ParentC. Response1: ParentC withdrew from the study after three weeks of enrolment. Consequently, their data is missing from the analysis, but we have maintained the original pseudonyms for consistency and clarity in the study documentation. We have added this information in ‘Participants’ section (Line between 120-123).
Comments3: Subsection 2.4 Intervention Materials should be placed before subsection 2.3 Procedure, that discusses the phases of the study, before those are even described. Response3: We agree with your suggestion to place Subsection 2.4 Intervention Materials before Subsection 2.3 Procedure, as it provides a more logical flow by describing the intervention materials before discussing the study phases. We have revised the manuscript accordingly and now the Intervention Materials section comes before the Procedure section.
Comments4: Method section needs a bit more care Response4: We have made adjustments in methods section accordingly. |
||
|
Additional clarifications |
||
|
We are open to any additional suggestions you might have. Thank you once again for your constructive feedback. |

Round 2
Reviewer 2 Report
Comments and Suggestions for Authors
All the revisions have been made.
Author Response
Dear Reviewer,
We are pleased to receive confirmation that all required revisions have been satisfactorily addressed. We appreciate the time and effort taken to evaluate our manuscript and provide constructive feedback. We have also incorporated the feedback provided by the academic editor.
Thank you for the opportunity to improve our work.
Dr Fatma Canan Durgungoz
Dr Michelle C. St Clair